# The Unique Experience of Intersectional Stigma and Racism for Aboriginal and Torres Strait Islander People Who Inject Drugs, and Its Effect on Healthcare and Harm Reduction Service Access

**DOI:** 10.3390/ijerph22071120

**Published:** 2025-07-16

**Authors:** Emily Pegler, Gail Garvey, Lisa Fitzgerald, Amanda Kvassay, Nik Alexander, Geoff Davey, Diane Rowling, Andrew Smirnov

**Affiliations:** 1School of Public Health, University of Queensland, Herston, QLD 4006, Australia; g.garvey@uq.edu.au (G.G.); l.fitzgerald@uq.edu.au (L.F.); a.smirnov@uq.edu.au (A.S.); 2Queensland Injectors Health Network (QuIHN), Bowen Hills, QLD 4006, Australia; akvassay@quihn.org (A.K.); nalexander@quihn.org (N.A.);; 3Metro North Sexual Health and HIV Unit Windsor, Brisbane City, QLD 4032, Australia; diane.rowling@health.qld.gov.au

**Keywords:** harm reduction, injecting drug use, Indigenous peoples, lived/living experience, stigma, racism, healthcare service access

## Abstract

Aboriginal and Torres Strait Islander people who inject drugs face persistent health inequities, highlighting the need for programs that meet the needs of these groups. This study explored how intersectional stigma and discrimination affect Aboriginal and Torres Strait Islander people’s access to quality healthcare. Aboriginal and Torres Strait Islander participants aged ≥18 years who had injected drugs within the past 12 months were recruited from two regional needle and syringe programs (NSPs) and a major city NSP in Queensland, Australia. Participants completed a structured survey and yarned with an Aboriginal researcher and non-Indigenous research assistant about their healthcare experiences. Through a process of reflexive and thematic analysis, three major qualitative themes emerged: participants’ social circumstances and mental health challenges made help-seeking difficult and complex; enacted stigma and racism diminished access to health services and the quality of care received; and injecting drug use was associated with disconnection from culture and community. Privileging the expertise and voices of those with lived/living experience is essential for the creation of culturally safe, inclusive, and destigmatising healthcare services for Aboriginal and Torres Strait Islander people who inject drugs.

## 1. Introduction

Stigma and racism in healthcare systems have being pivotal in establishing pervasive health inequities for Aboriginal and Torres Strait Islander peoples, significantly informing their healthcare seeking decisions, health behaviours, and undermining their physical, cultural and emotional wellbeing [1,2].

Stigma describes a social process characterised by exclusion and discrimination based on an attribute or health behaviour that positions an individual outside the margins of society [3,4,5]. Drug use stigma is mobilised through the exercise of power, reflecting a social consensus that devalues people who use drugs, and legitimises collective action to criminalise this population through healthcare systems and punitive policies and practices [6,7]. People who inject drugs (PWID) are perceived by sections of the society as dangerous, unpredictable, and inclined to commit acts of violence and crime [5,8,9,10], and injecting drug use is commonly associated with other socially stigmatised health conditions, such as, HIV, hepatitis C and mental health issues [8,11,12].

There is a substantial body of research indicating that experiencing stigma and racism contributes to adverse health and wellbeing outcomes for people engaged in drug use, including injecting drug use [10,13,14,15]. Injecting drug use-related stigma has been linked to increased participation in risk practices, such as shared use of injecting equipment [13,16] and sexual risk behaviours (e.g., unprotected sex) [16,17], thus contributing to high infection and transmission rates of blood-borne viruses (BBV) and sexually transmissible infections (STI) [16] and experiences of poor mental health and emotional wellbeing [10,11,18]. Stigma and racism also create dangerous barriers for PWID who seek lifesaving harm reduction services [8,15,16,19]. The consequences of these barriers include lower utilisation of needle and syringe programs (NSPs) [20], the development of strategies to conceal evidence of drug use, and downplaying pain [8]. There is limited evidence investigating the confluence of stigma and racism on health equity and wellbeing for Aboriginal and Torres Strait Islander PWID, who are at the intersection of densely woven patterns of systemic injustice [10,21,22].

Stigma operates on both interpersonal (enacted, anticipated, perceived and internalised) and structural levels [4,5,10,23,24]. Enacted stigma encompasses behaviour and practices that impose disadvantage, racism, or prejudice towards a stigmatised group [6,18,22]. Anticipated stigma describes the awareness of societal attitudes and past experiences of stigma that elicit fear and anticipation of experiencing stigma again in the future [6]. Perceived stigma refers to the belief held by a stigmatised person that society as a whole ascribes to the stereotypes of people from that group [10], and internalised stigma refers to when individuals believe stigmatising attributions about themselves [20,25,26]. On a structural level, stigma upholds systems of oppression whereby those in positions of power maintain the status quo by demoralising people who use drugs, and blaming stigmatised individuals or groups for their sufferings [10,23]. For example, a person engaged in injecting drug use may be criminalised and incarcerated and obtain a criminal record, and as a method of “keeping them down”, their job opportunities may decrease or they may face barriers finding stable and affordable housing [5,24]. Drug use-related stigma is also a reflection of social hierarchies which operate within processes of health service governance and service design [27]. Despite growing efforts to include people with lived/living experience in service design and delivery, these voices have historically been excluded from these spaces and positions of power due to being labelled as hard-to-reach or difficult to work with [27].

Racism and stigma are inextricably linked and function at a systems level to exploit, dominate and enforce social norms [24]. This aligns with Feagin’s theories on racism, whereby racism is rooted in the “system” and in social institutions rather than the individual [28]. Stigma and racism experienced by Aboriginal and Torres Strait Islander people can also be explained by these theories, which highlight the roles healthcare systems, government, colonialism and cultural ideologies play in perpetuating racism and stigma for these groups [5]. Systemic and institutional racism highlight the ways in which colonialism legacy has perpetuated the devaluation of Aboriginal and Torres Strait Islander people [29], and ultimately create intersectional experiences and inequities to accessing healthcare, including harm reduction services [19,29]. Intersectionality and intersectional stigma theories posit that when multiple sources of stigmatisation of a group co-occur, they coalesce to form a new set of experiences of discrimination for those who reside at the intersection. It is important to consider how racism and stigma, associated with injecting drug use, Indigeneity, gender, sexuality, or other aspects of identity, simultaneously interact in varied ways to produce health inequities.

For Aboriginal and Torres Strait Islander people, stigma and intersectionality have been explored for some topics, such as methamphetamine use [30], healthcare access and treatment of other stigmatised conditions (e.g., HIV, hepatitis C, mental illness) [11,15,25,26,31,32,33], oral health [34], incarceration [35] and disability [36,37]. Yet there has been little exploration of the synchronistic influence and coalescence of multiple forms of stigma for Aboriginal and Torres Strait Islander PWID.

Harm reduction and other healthcare services for PWID have not integrated Aboriginal and Torres Strait Islander cultures and ways of knowing and being into the design and delivery of services [38], and there is extensive evidence that Aboriginal and Torres Strait Islander people are not receiving culturally safe and equitable care when it is needed and that mainstream health services do not always consider the holistic concept of Aboriginal and Torres Strait Islander health wellbeing, which is centred on relationality encompassing social, spiritual, physical and cultural determinants [39,40,41]. The Social and Emotional Wellbeing Framework developed by Gee et al. [42] reflects a holistic view of health considering that the wellbeing of Aboriginal and Torres Strait Islander people is shaped by connection to culture, kinship, community, and Country, which are all deeply reciprocal and linked to identity, health and healing [40]. The framework also considers the social, political, historical factors that have contributed to the stigmatisation and marginalisation of Aboriginal and Torres Strait Islander PWID [42]. Evidence from other population groups and settings suggest that holistic, culturally safe and non-judgemental healthcare can promote client engagement with services and improve treatment retention and health outcomes [9,33].

We have applied intersectionality theories to understand the multiple forms of stigma and discrimination experienced by Aboriginal and Torres Strait Islander PWID in Meanjin (Brisbane) and the regional towns of Gimuy (Cairns) and Gurambilbarra (Townsville). We explore how these experiences may impact or limit access to harm reduction and other healthcare services and the quality of care obtained at these services.

## 2. Materials and Methods

### 2.1. Design and Ethics

This study is part of a project funded by the Sexual Health Ministerial Advisory Committee (SHMAC) Research Fund. Ethical approval for this project was obtained from the University of Queensland Human Research Ethics Committee (approval number: 2021/HE000742). The research team comprised Aboriginal and non-Aboriginal researchers and investigators within and outside of UQ, including a research coordinator who is a proud descendant of the Yamatji people. The project coordinator and stakeholders brought extensive expertise to the research design and process, including knowledge of racism and the impacts of colonisation, as well as culturally informed research skills such as research interview yarning.

This project took steps to establish equal research partnerships with Aboriginal and Torres Strait Islander organisations and stakeholders to uphold and honour cultural security, governance, and meaningful engagement with communities. The research team carried out extensive preliminary consultations with our research partner, the Queensland Aboriginal and Islander Health Council (QAIHC). QAIHC and other stakeholders had an active role in project planning, including research design and establishment of fieldwork sites and processes, and provided cultural expertise on resources such as the Terms of Reference for Aboriginal and Torres Strait Islander consumers. These governance processes aligned with the CONSolIDated critERia for Strengthening the Reporting of Health Research Involving Indigenous Peoples (CONSIDER) statement which was developed to advise researchers on best-practice approaches to conducting research with Aboriginal and Torres Strait Islander people [43].

### 2.2. Sampling and Recruitment

Participants included Aboriginal and Torres Strait Islander PWID who were over the age of 18 and had injected within the past 12 months. Participants were recruited at and referred to the needle and syringe program (NSP) research sites located at QuIHN (Meanjin/Brisbane and Gurambilbarra/Townsville) and Youth Link (Gimuy). These organisations operate not-for-profit primary (i.e., dedicated) NSPs and other harm reduction, counselling and support services. Participants were recruited using location-based convenience sampling and took part in point-of-care testing (POCT) for a range of BBVs and STIs, a supported quantitative survey, and semi-structured qualitative interview yarns. NSP staff at QuIHN and Youth Link informed eligible clients about the project. Participants were offered a AUD 40 voucher for completing the POCT and survey, and a AUD 20 unconditional pre-paid peer recruitment incentive was offered to participants to refer a friend or family member to the research. Relying on the principle of reciprocity, participants received this voucher regardless of whether their referral participated. Participants who completed an interview were offered an additional AUD 40 voucher.

### 2.3. Survey Measures of Stigma and Discrimination

#### 2.3.1. Discrimination and Racism Experienced by Aboriginal and Torres Strait Islander People in Everyday and Healthcare Settings

Everyday discrimination and healthcare discrimination were measured using scales developed and validated in the “Mayi Kuwayu: The National Study of Aboriginal and Torres Strait Islander Wellbeing”, a large prospective community study [44]. The scales were adapted from the Everyday Discrimination Scale [45] and other measures, which were modified and refined through focus groups, field testing, and input from stakeholders with lived experience and/or content expertise. Participants were asked how often a range of things happened to them (everyday discrimination: 8 items; healthcare discrimination: 4 items) with the response options “not at all” (coded as 0), “a little bit” (1), “a fair bit” (2), and “a lot” (3). Item responses were summed to create a total for everyday discrimination (range: 0–24) and healthcare discrimination (range: 0–12). For each scale, “any discrimination” included all observations with a total score ≥ 1. “High-level” discrimination” was categorised according to a cut-off of 17 for the Everyday Discrimination Scale and 9 for the Healthcare Discrimination Scale [44]. Principal Axis Factor analysis results were consistent with the two distinct factors of everyday and healthcare discrimination. The Everyday Discrimination and Healthcare Discrimination Scales each had good internal consistency (Cronbach’s alpha of 0.894 and 0.853, respectively). Both scales had strong convergent validity with a single-item question on community racism (asking participants whether racism is a problem where they live). In addition, a single-item attribution question was developed to follow each scale (“When these things happen, do you think it is because you are Aboriginal/Torres Strait Islander?”) with the same response options as the other items.

#### 2.3.2. Stigma Associated with Substance Use

Stigma associated with substance use was measured using the Substance Use Stigma Mechanisms Scale [6]. The development of the SU-SMS was guided by the Stigma Framework, which was informed by stigma research across disciplines and identifies the dimensions of enacted, anticipated, and internalised stigma [46]. The SU-SMS measures enacted stigma in family and healthcare settings (3 items for each setting) and, similarly, anticipated stigma in family and healthcare settings (3 items each). There are a further 6 items measuring internalised stigma. Levels of agreement with items (e.g., “Having used alcohol and/or drugs makes me feel like I’m a bad person”) were assessed using a 4-point scale: “not at all” (0), “a little bit” (1), a fair bit (2), and “a lot” (3). Mean scores were calculated for each scale (range: 0–3). A cut-off of 1 was used to indicate “any” stigma and a cut-off of 2 was used for “high-level” stigma in all scales. These measures have been validated in culturally diverse US samples of opioid dependence treatment and HIV clinical care patients, including initial pilot testing involving cognitive testing with these groups. Confirmatory factor analysis indicated that all items load on their hypothesised constructs. All scales (alpha = 0.90–0.93) and subscales (alpha = 0.90–0.95) have high internal consistency.

#### 2.3.3. Stigma Associated with Hepatitis C

We used the Hepatitis C Virus (HCV) Stigma Scale, which was developed by Cabrera [47] and was an adaptation of the 10-item HIV Stigma Scale developed by Wright et al. [48]. Wright et al. [48] examined the items of the original 40-item HIV Stigma Scale developed by Berger et al. [49] and selected items that loaded highest on each of the subscales of the original scale, which resulted in 10 items distributed across the subscales: personalised stigma (3 items), disclosure concerns (2 items), negative self-image (3 items), and concern with public attitudes (2 items). Cabrera [47]) directly adapted the items identified by Wright et al. [48]; e.g., “Having HIV makes me feel that I’m a bad person” was changed to “Having hepatitis C makes me feel that I’m a bad person”. Responses were coded on a 4-point scale, “Strongly Disagree” (1), “Disagree” (2), “Agree” (3), and “Strongly agree” (4). Mean scores for each scale were calculated and a cut-off of 3 was used to indicate “any” stigma, with a cut-off of 4 for “high-level” stigma. For this study, we only used the ‘negative self-image’ and ‘concern with public attitudes’ subscales, which are closely related with the Stigma Framework constructs of internalised stigma and anticipated stigma, respectively. For consistency within this study, we use the Stigma Framework terms when referring to these subscales. The HCV Stigma Scale has shown a high level of internal consistency (alpha = 0.86–0.92). There is also high correlation between individual items and the total scale (0.55–0.80). A positive correlation between stigma scores and depressive symptoms reported on the Center for Epidemiological Studies Depression Scale (CES-D; r = 0.43, *p* = 0.01) is suggestive of convergent validity.

### 2.4. Data Collection and Yarning Process

Each interview yarn began with a social yarn (a relational Indigenous methodology of two-way knowledge exchange) to cultivate rapport and show authentic interest in the participant’s life [50]. During the social yarn, it was important for the interviewer to share details of who their mob are, and why they are involved in this research, and then hold space for the participant to share their kinship ties and any feelings about the interview. This exchange meant the participant’s cultural identity and selfhood was seen, respected and safeguarded by the researcher. When conducting research with Aboriginal and Torres Strait Islander people, researchers reflected on how our emotions and privileges impacted the ability to develop trust with the participants, especially when positioned within health services, which for Aboriginal and Torres Strait Islander people can be culturally unsafe and hold memories of neglect and trauma.

Interview length varied, on average lasting approximately forty minutes. Questions in the interview guide included five broad topic areas and a range of prompts. Questions began with place-based and demographic questions such as “how do you feel in this community?” and “What makes you feel strong?”, followed by questions about experiences in mainstream healthcare services: “have you ever been made to feel unwelcome/welcome in any healthcare setting?” and “Do you feel as though you are treated the same as non-Aboriginal and Torres Strait Islander clients?”. The next section covered experiences with harm education services, including NSPs at QuIHN and Youth Link: “how often do you use NSPs?” and “Have your experience with NSPs affect your decisions to get treated for a viral and bacterial infection, such as hepatitis C…?”. Interviews were recorded and fully transcribed by a local external transcribing service. Names and identifiable information were removed from transcription.

### 2.5. Data Analysis

#### 2.5.1. Quantitative Data

The survey data were analysed descriptively (e.g., mean, proportions, confidence intervals). Data was cleaned and analysed using Stata SE version 18. De-identified (quantitative and qualitative) data were stored in the university’s secure data management folders with access restricted to the research team.

#### 2.5.2. Qualitative Data

Qualitative interview data were analysed reflexivity and thematically by the Aboriginal research coordinator, with involvement and guidance from Aboriginal advisors, stakeholders and non-Indigenous members of the research team. Interview transcripts were transcribed verbatim, checked for accuracy, and imported into NVivo (v14, QSR, MA, USA), a qualitative analysis software program where a codebook was developed. Analysis was conducted using deductive and inductive processes following Braun and Clarke’s [51] recommended six steps of thematic analysis (data familiarisation, generating initial codes, searching for themes, reviewing themes, defining and naming themes, interpretation and reporting). Relevant features of the data (both semantic and latent) were read carefully and coded for aspects relating to the intersectional experiences of stigma for Aboriginal and Torres Strait Islander PWID, including the impact of stigma and other vulnerabilities on harm reduction and health service access. Induction facilitated the identification of major themes in the dataset, some of which corresponded to intersectional stigma, describing issues of drug dependence, cultural identity, racism, and the role of family and community. Deductive analysis was guided by intersectional stigma theory and previous research on the intersectional experiences of stigma for people engaged in substance use, including barriers to accessing harm reduction and health services, institutional racism, and the impact of mental health and social vulnerabilities on health equity. Aboriginal and Torres Strait Islander worldviews and ways of knowing and being are embodied by relationality and intwined with social and ancestral connections to Country, culture and kin [37]. The research team engaged in reflexive discussions throughout analysis, examining how our inherent biases and positions might influence interpretations. For example, the research coordinator acknowledged themselves as an “outsider”, in both a relational and physical sense, having no personal experience or connection with being an Aboriginal and Torres Strait Islander person who injects drugs, and having never accessed harm reduction services.

## 3. Results

### 3.1. Survey Findings: Social and Demographic Characteristics and Measures of Stigma and Discrimination

Of the 94 participants who completed the survey, 92 (98%) completed questions about their social and demographic characteristics. The following estimates of participant characteristics are based on this group (*n* = 92). A majority of participants were men (65%), 32% were women, and 3% were non-binary or preferred not to say. The mean age of participants was 41 years (range 22 to 63 years). With regard to sexuality, 87% of participants were ‘straight’/heterosexual, 10% were bisexual, and 3% identified as ‘other’ or preferred not to say. The highest level of education completed was primary school for 11%, junior secondary (Year 10 or equivalent) for 45%, senior secondary (Year 12 or equivalent) for 25%, a certificate/diploma for 8%, and a bachelor’s or postgraduate degree for 8% of participants. A further 3% reported not completing any schooling. With regard to current accommodation, 19% reported living in a boarding house or hostel, 18% were in a private rental (including shared rental accommodation), 17% were in public/community housing, 17% were ‘sleeping rough’ (on the street, in parks, and in makeshift shelter) or living in a squat, 9% shifted between their family’s, friend’s or acquaintance’s places, 2% lived in a home they owned, and 15% reported ‘other’ accommodation. A total of 19% of participants reported that they had ever been in youth detention, and 57% reported ever having been imprisoned.

A total of 94 participants completed the survey questions on stigma and discrimination. Questions in the survey pertained to experiences of discrimination in healthcare environments, everyday life, with family or friends, and in relation to hepatitis C infection. The measures on stigma and discrimination in Table 1 show that a large proportion of participants had experienced at least some level of everyday or healthcare discrimination, enacted substance use stigma from family and friends, and internalised stigma and shame related to their substance use. For those who had been infected with hepatitis C, stigma associated with this condition was also common. A large majority (85%) had experienced some everyday discrimination (e.g., ‘I am treated with less respect than other people’; ‘People act like they are afraid of me’) and also a majority had experienced discrimination in healthcare settings (e.g., ‘Health care providers do not listen to what I say’; ‘I go home without the care I need’). Regarding attribution, 53.2% (*n* = 94) thought that the everyday discrimination occurred because they are Aboriginal and/or Torres Strait Islander, and for healthcare discrimination, 41.5% thought this was the case.

### 3.2. Qualitative Interview Yarn Data

Interview yarns were conducted with 14 participants, both individually and some as couples. All participants identified as Aboriginal and/or Torres Strait Islander and 64% (N = 9) were male. Three major themes emerged from the data: social circumstances and mental health exacerbate drug dependence and make seeking support services challenging and complex; enacted stigma and racism diminish access to health services and the quality of care received; and injecting drug use is associated with disconnection from cultural identity and community.

#### 3.2.1. Social Circumstances and Mental Health Exacerbate Drug Dependence and Make Seeking Support Services Challenging and Complex

Participants described the social circumstances and mental health issues they experienced and how they created challenges in accessing health and social care services. Social circumstances included homelessness and unstable housing, incarceration, food and financial insecurity, lack of affordable transportation, and family issues. When asked about what they might change about their life, one participant shared the need to address their housing insecurity issues:


*Probably the main thing I need to do is find a longer-term suitable living place… I’d rather have the housing, housing, apartment unit where there’s some lower level closer. Closer back out to my friends…or even near a public transport*
(Participant 4, male, 50s)

Social circumstances and generational traumas had a profound impact on drug dependence. Drug use intensified poor mental health, which in turn meant that participants were less motivated to engage with health services. One participant shared that their drug use acted as a coping mechanism and emotional defence against the pain and trauma that was inflicted upon him as a child. Another participant shared the all-consuming and burdensome impact of drug use on his life, which depleted his motivation to access and engage with health services:


*It’s a never-ending battle to try and get drugs, then it’s a never-ending battle to try and get off drugs. I don’t know, it consumes all your time, you’ve got nothing really left after that. It’s hard to engage and do anything else, if you can’t 100 per cent something else then it’s pointless really, isn’t it? Nobody really wants to give you a shot anyway because they know that you’re just a waste of fucking time*
(Participant 13, male 40s)

Participants discussed the complex and onerous challenges of navigating health and social services including crisis accommodation and rehabilitation services. In one regional site, participants spoke to the intersections of substance use discrimination from services and broader society. Two participants described feelings of rejection and unfairness when attempting to access a homeless shelter that only offered support for alcohol users:


*They have a sober shelter but there’s only one of those as well and there’s no real homeless centre for drug users, where they’re not discriminated by drawing the line between the different drug or alcohol that they use*
(Participant 7, male, 40s)

Accessing services, including homeless centres, was a compelling issue for women engaged in drug use, who faced added safety concerns and gendered violence. One participant discussed safety concerns for their female partner sleeping rough:


*Yeah, or for some of the women……they’ve got somewhere to go to sleep…. She’s worse, you’re making her sleep in the street, not only can she not hide…. she doesn’t hear anything that goes on around her, you know what I mean?*
(Participant 7, male 40s)

Participants identified the barriers to finding safe and affordable housing without consistent rental history, especially when having experienced incarceration. Such hurdles to entering the rental market keep people who use substances in vicious cycles of financial hardship and homelessness. One participant shared that


*A lot of people don’t have rental history to even get their foot in the door with any real estate. A lot of the hotels and stuff too rely on the rental history as well”. They said, “I find that’s why there are a lot of drug users couch surfing or making camps in the bush and stuff like that*
(Participant 6, female 30s)

#### 3.2.2. Enacted Stigma and Racism Diminish Access to Health Services and the Quality of Care Received

Aboriginal and Torres Strait Islander PWID often and repeatedly felt unsafe, undervalued and markedly stigmatised at health and hospital services. Participants shared their experiences of stigma and racism in their interactions with health services and staff. Their narratives described complex intersectional experiences of navigating health services and being unable to distinguish what they were being stigmatised against—Indigeneity, drug use, age, gender, or a combination of all these factors.

Participants experienced enacted stigma and discrimination from hospital services, including the emergency departments. Many participants provided examples of receiving poor-quality care and being subjected to derogatory labelling when accessing hospital services due to their status as a person who injects drugs:


*Well, when we’re going to a hospital. Yeah. You know, they treat us like shit…especially if you get dragged in on an overdose or an episode. Yeah, from drugs or whatever… they label as junkies, and they’ve even said it to my face. Anyone who goes up to the hospital and, um, my intravenous users, they’re just, they get the poorest quality care*
(Participant 1, female 40s)

Participants shared both positive and negative experiences at NSP sites but described feeling safe and welcomed at peer-led services which were kind, empathetic and knowledgeable. Settings where participants felt judged and unwelcomed included secondary NSPs (operating within mainstream health services) and government-run NSPs. Other health services where participants repeatedly felt disrespected and devalued included pharmacies and general practitioners (GPs). Participants spoke about acquiring injecting equipment from local pharmacies and being subjected to insulting looks. At GP practices, participants emphasised feeling unheard, misjudged, and found it tiresome trying to prove their health concerns were valid. One participant shared that


*You’re trying to say, listen, I need some more Valium…because we present ourselves as we’re doing, we seem to be doing all right…. You go the doctor’s appointment with them, and they go, you seem to be doing all right. And it’s like I wouldn’t be sitting here if I was as doing really well *
(Participant 1, female 40s)

One participant, when describing his experience of feeling unwelcome when accessing a pharmacy to acquire injecting equipment, described efforts to be discrete about his drug use out of fear of stigmatisation from the staff and public:


*You’re standing there with normal people you know what I mean? You’re trying to be a little bit private and discreet about it. Yeah, they’re just looking at you like mmmm what are you doing you know what I mean?*
(Participant 3, male 30s)

Several participants believed that noticeable evidence of drug use diminished their access to quality care and respect. Participants spoke about efforts to present themselves as “clean” and “tidy” to influence positive perceptions about their character and intentions when accessing health services. This stemmed from an overarching belief that if they “look like a drug user” they will be susceptible to stigmatisation from staff. One participant expressed this belief, by sharing that they like to dress “presentable” when attending doctors’ appointments to conceal evidence of their drug use. By doing this, they believe they will receive respect and avoid any anticipated judgement or stigma.


*I walk around dressed up, I keep myself clean and tidy so no one will know I’m a user. That’s why try tell (unclear) keep yourself tidy so no one knows that you’re a user. Just dress up and look tidy a bit, and you will get respect*
(Participant 9, female 50s)

Experiencing disrespect from health staff discouraged participants to access services where they anticipated further stigma might occur.


*The hospital’s a prick too because they’ve got their vending machine right in front of the taxi rank, the police pull up right there… They’re giving you looks, you know what I mean…It’s like a little bloody cop farm there, so people are discouraged there*
(Participant 7, male 40s)

Feelings of internalised shame and stigma coincided with perceived stigma, where participants believed the public held negative views and attitudes about their drug use. For example, one participant spoke about acquiring injecting equipment from the vending machine outside at the emergency department and avoiding the area when families are around because of the shame they had internalised about their drug use.


*But sometimes I (feel) shame because there’s families around then I go, come back for an hour time*
(Participant 12, male 30s)

Participant narratives also revealed resistance to internalised stigma and shame, with descriptions of value and worthiness despite drug use and/or incarceration. Some participants felt their self-worth and self-respect derived from their strong moral compass and character, which acted as a foundation from which they derived strength, resilience and hope.


*Every day I’m given the opportunity to wake up and breathe and walk and drift and just be here…. it’s about learning and speaking and improving the way I was from yesterday or from three months… Just seeing how I’ve grown, and I’ve gone up and down…. We’re all human at the end of the day, so we’ve got to make mistakes. We’re going to make slip ups of course*
(Participant 5, male 40s)

Participants shared varied experiences about navigating health services as an Aboriginal and/or Torres Strait Islander person and the complex interrelationship of drug use, racial and gender stigmatisation. Many experienced, and observed, racism and disrespect from healthcare staff and doctors. In contrast, some participants described not experiencing racism and being treated with the same level of respect as non-Indigenous people because they did not “look” Aboriginal and felt their physical attributes changed the way healthcare staff perceived them, “*Especially because we’re white blackfellas*” (Participant 1, male 40s). For others, disclosing their Indigeneity in health services engendered feelings of fear and anticipated racism and mistreatment from staff.


*I think sometimes if once they find out that you’re ABSTI (Aboriginal and Torres Strait Islander) they might treat you differently or they might look at you differently”*
(participant 5, male 40s)

Some acknowledged the ongoing history of institutional racism and mistreatment from staff against Aboriginal and Torres Strait Islander people in Australia. Participants alluded to the unchanging and enduring mistreatment of mob who access health services that are not culturally safe.


*It’s just the way it is, it’s the way people are. It’s been happening for years, and it was never just going to stop, and everyone was going to shake hands and be friendly. There’s always been that fucking bullshit there, you know and there always will be… it’s easier to pick on a black man, isn’t it? We’ve always been picked on*
(Participant 13, male 40s)

Internal narratives from participants reflected differing perceptions regarding what aspects of their identity they were being stigmatised or discriminated against. Many perceived that discrimination, racism or stigmatisation they experienced was partially due to their drug or substance use, more than their Indigeneity. Other participants acknowledged the multiple layers and intersections of stigma they experienced when accessing a local hospital.


*“I think it may be all wrapped in one, Aboriginal and Torres Strait Islander and female” (Participant 6, female 30s), and their partner saying, “I think it’s more the stigma, just being a user”*
(Participant 7, male 40s)

#### 3.2.3. Injecting Drug Use Is Associated with Disconnection from Cultural Identity and Community

Participant yarns revealed poignant truths about navigating cultural identity, performing cultural responsibilities and the complex feelings of connecting with family, community and Country whilst being engaged in drug use. Many participants described feeling shamed, ostracised, and disconnected from Country and community because of their drug use, and the profound effect this had on their willingness and motivation to access health and harm reduction services. Cultural attitudes and negative perceptions of drug and alcohol use within communities meant that for among many Aboriginal and Torres Strait Islander communities, any substance use was discouraged, and this was enforced by the leadership of Elders.


*We don’t even let anybody take it (drugs) up there. If we find out that person is going to get kicked…*
(Participant 12, male 30s)

Many participants shared examples of feeling discouraged and afraid to access a health and harm reduction service to acquire injecting equipment due to the risk of being recognised by friends or family who worked at a service and being exposed to carry-yarn (community gossip). When exploring solutions to make harm reduction services more culturally safe and welcoming, many participants believed that hiring more Aboriginal and Torres Strait Islander staff could potentially lead to stigmatising and unsafe experiences.


*Yeah, yarn carriers. They don’t keep a secret, (unclear) the whole thing that they just- and they give you a look, like, (a freaky look). She’s got something. Oh, no, they’re terrible, they’ll judge us*
(Participant 9, female 50s)

A participant echoed this by describing the shame that many Aboriginal and Torres Strait Islander PWID feel, and the burden of concealing drug use when accessing a service with Indigenous staff.


*A lot of them might feel ashamed about it or they don’t want to talk about it to actual other Indigenous people, which I totally get. Yeah. Um, I myself, I’m of that opinion. There are conversations that I keep separate from when I go to the service like this. Yeah. I’d rather have a, um, confidential discussion with our doctor*
(Participant 5, male 40s)

In addition to feeling discouraged when accessing health and harm reduction services staffed with Aboriginal and Torres Strait Islander people, participants spoke about experiences of being kicked out, ostracised and shamed by their communities due to their drug use.


*You’ve got the elders that they find disrespectful that their kids are using anyway. You know what I mean? Like, it’s hard as an Aboriginal to fucking use because. Yeah. Because so much shame. Yeah. I mean, the elders that shame, they’ll push you out of the family*
(Participant 3, male 30s)

One participant yarned about the duality and intersections of their identity, where he kept his drug use and his cultural identity and responsibilities separate and sacred to the “character” he embodies.


*When I’m back home I’m a different person…I like my title, my role. I’m that person back home. When I’m here I’m different too so there’s two characters (here)*
(Participant 12, male 30s)

Several participants expressed narratives of resilience in the face of stigmatisation, and how actively resisting stigma is a form of healing and reconnecting cultural identity and belonging. One participant shared how he embraces his innate and enduring strength as a survivor and warrior:


*I’m—that’s what I say I’m tired of people (carry-yarn) me like why the Torres Strait talks about me, they’ll carry on me. That thing has been happening for years, so I don’t worry now like if they want to say anything they say it to my face…it doesn’t faze me, I don’t give a shit. I’m from the bushes. I can survive. I’m a survivor. I’m a warrior”*
(Participant 12, male 30s)

Participant narratives also revealed that community and family were an unwavering pillar of support that they derive strength, community and sense of belong from. Others felt a sense of kinship responsibility and accountability to help community members and mentor and protect younger mob from engaging in substance use. When asked how they would like to feel more engaged in the community one participant replied


*Like helping them. Helping them young kids, you know, in the community…just helping them, because I know about it, and tell them about it, too…I know about drugs, and when I see young kids using, I don’t feel right that they’re doing it. It mucks you up*
(Participant 9, female 50s)

## 4. Discussion

Our study conveys nuanced participant accounts of distressing and disruptive experiences of stigmatisation in healthcare and harm reduction service settings. The survey findings indicate that participants had experienced high levels of everyday discrimination, healthcare discrimination, internalised stigma and shame, and stigmatisation from friends and family. Our interview yarns identified three prevailing themes encompassing participant experiences of stigma and racism across a variety of health and harm reduction service settings: (1) social circumstances and mental health exacerbate drug dependence and make seeking support services challenging and complex, (2) enacted stigma and racism diminish access to health services and the quality of care received, and (3) injecting drug use is associated with disconnection from cultural identity and community. These themes provide impetus to progress towards transforming health services and systems for Aboriginal Torres Strait Islander PWID, re-centring them on culture, inclusivity and healing, whilst dismantling racist colonial structures. Applying an intersectional lens to this research has highlighted the complex interplay of group stigmatisation and the intersecting barriers to seeking and accessing health services that are amplified by the culmination of stigma, racism, social vulnerability and poor health.

Our study appeared to find higher proportions experiencing everyday discrimination and healthcare discrimination compared to the Mayi Kuwaya Study [44], which was a large community study of Aboriginal and Torres Strait Islander peoples. In the current study, 85 per cent experienced everyday discrimination and 70 per cent experienced healthcare discrimination, compared with 55 per cent and 34 per cent in the Mayi Kuwaya Study. These findings suggest that experiences of discrimination, including racial discrimination, are particularly common for Aboriginal and Torres Strait Islander PWID, and that these levels of discrimination are experienced in a range of everyday community settings and in healthcare settings. Similarly, our findings for substance use stigma also indicate that participants were commonly exposed to stigma and discrimination from different sources, including stigma from family and within healthcare settings. Further, for those participants who had been hepatitis C-positive, many experienced stigma associated with this condition. The findings of our interview yarns support and elucidate these survey findings. The interview yarn findings revealed multiple accounts of enacted, internalised and anticipated stigma, which is reflected in the survey findings. For example, participants spoke about stigma and discrimination in mainstream health settings, including hospital emergency departments, and some participants shared experiences of anticipating stigma from family members and community. These findings of multiple forms of stigma and discrimination highlight the pervasive and intersectional nature of these experiences for Aboriginal and Torres Strait Islander PWID.

Our findings suggest that the intersection of stigma and situational factors such as homelessness has a pervasive role in the health-seeking experiences of Aboriginal and Torres Strait Islander PWID, which is consistent with the findings of recent studies of PWID in other population settings [15,23]. Experiences such as homelessness and drug dependence were described as burdensome and all-consuming and made participants feel like good health and wellbeing were not deserved. This was reinforced by enacted and perceived stigma from healthcare staff and some public attitudes towards people engaged in drug use. These narratives are consistent with previous work describing how situational factors and drug use combine to create “societal stigma”, and the idea that stigmatised groups may feel less motivated to seek care and treatment due to internalised feelings of shame, low self-worth and the belief that help and support are not deserved [15,52]. These experiences align with Graham Scrambler’s [4] theory that when a person is stigmatised at an individual and structural level, they lose status in society, leading them into “a spiraling of disadvantage”. Participants’ experiences reflect this cycle. Multiple manifestations of stigma (from drug use, racial identity, and homelessness) were internalised, present in healthcare interactions, and reinforced by larger societal institutions and systems that “kept people down” through barriers such as difficulty acquiring rental history.

Aboriginal and Torres Strait Islander participants were manifestly stigmatised and experienced racism at healthcare services, particularly in hospital settings, when attending GPs, and at pharmacies. Participant narratives revealed distressing truths about feeling unsafe, unheard, and neglected across multiple levels and settings in healthcare, due to racist attitudes, and derogatory labelling by healthcare workers. For participants, various types of stigmas were interconnected, and together they contributed to internalised feelings of shame about drug use, distrust of healthcare workers, and avoidance of seeking healthcare. Previous studies have shown that many Aboriginal and Torres Strait Islander people are likely to distrust and avoid health and hospital services, including those staffed by non-Indigenous healthcare workers, because they represent authority and enduring colonial powers [53]. These experiences are exacerbated by stigma associated with drug use and other personal characteristics including gender and sexuality. Stigma and discrimination in healthcare settings can translate to poor health-related outcomes and contribute to symptoms of stress, depression and anxiety, less frequent visits to health providers, and lower adherence to health regimes [16,54,55]. This can lead to an increase in negative health behaviours and risk taking such sharing injecting equipment, leading to higher instances of BBVs [13], and increased risky sexual behaviours [17].

Our analyses showed that some PWID believe that noticeable evidence of drug use attracts stigmatisation and diminished respect from healthcare workers. According to Erving Goffman, people from stigmatised groups must continuously and consistently work to “maintain face” and present socially acceptable versions of themselves to fit into broader society [4,56] and are expected to reduce tensions in their encounters with others in public settings. This was reflected in accounts of participants avoiding hospital vending machines when acquiring injecting equipment in fear of being publicly visible. Participants believed that if they “looked like a drug user” (e.g., visible track marks) they would not receive respect, care and trust from healthcare workers who they perceived might accuse them of behaviours such as “pill shopping”.

Participants shared emotional and complicated truths about navigating cultural identity, performing cultural responsibilities and feeling disconnected from culture, family, community and Country whilst being engaged in drug use. Other studies reflect these experiences where healthcare workers have reported that Aboriginal and Torres Strait Islander people feel shame about their injecting drug use, fuelled by negative attitudes towards injecting drug use within communities [53]. A study by Treloar et al. [19] described Aboriginal and Torres Strait Islander peoples “automatic” and internalised expectations that their communities would have stigmatised health conditions. Participants also spoke to intersectional resilience and “warrior” identities, and actively resisting stigma as part of survival, healing and strength. Stigma resilience is crucial on an individual and community level and may contribute to redefining community norms and values and inviting healing practices and ceremonies in community [19,57]. Resisting stigma has been explored in other studies within Indigenous communities, including a study by Treloar et al. [19] in which participants recognised the stigma attached to hepatitis C and were actively campaigning against this in their communities.

### 4.1. Implications for Policy and Service Delivery

Our findings highlight the need for community-driven, culturally safe, and inclusive healthcare and harm reduction services for Aboriginal and Torres Strait Islander PWID that recognise holistic definitions of health and wellbeing and support the inclusion of strength-based approaches to service design and delivery [58].

Participants felt most safe and welcomed at primary needle and syringe programs, where they encountered staff with lived/living experience who have insight regarding the complex experiences and systemic adversity faced by PWID. This is supported by extensive evidence suggesting that primary healthcare [59], Aboriginal community-controlled health organisations, peer-led services [60], and culturally informed approaches to reduce stigma [58] are most accessible and acceptable for this population group [58,61].

Recognising culture, community and strength-based approaches, which are sources of healing and resilience and reduce the impact of shame, has imperative policy implications [58]. Harm reduction services operating from a holistic whole-of-health framework could mean integration of cultural and social support services and activities. Collective community-led strategies that prioritise building the resilience of PWID, offering programs that foster connection to culture and Country, and bringing people and communities together should be supported by policy and funding [40,58,62]. For example, connection to Country is vital to improving health equity and is shown to be a key contributor to positive health outcomes for Aboriginal and Torres Strait Islander people [40]. Cultural activities based on Country may reduce the impacts of stigmatisation and shame by building resilience and safeguarding cultural responsibilities and stewardship [40,58].

Furthermore, harm reduction services, alongside other community services, can play an integral role in supporting those who are experiencing homelessness through the implementation of integrative programs that are holistic, compassionate and incremental and centred on the principle of “meeting people where they are at” to address the intersectional barriers and intergenerational traumas experienced by these groups [63]. Several systematic reviews have examined the effectiveness of different types of interventions for socially marginalised PWID, including tailored primary care services [64] and the provision of housing, including Housing-First Models [63,65], which endorse the belief that people who are experiencing homelessness should be given immediate access to permanent housing and support that is individualised and rooted in harm reduction principles [63].

The involvement of Aboriginal and Torres Strait Islander people with lived/living experience and the leadership of Elders and community leaders in policy making, research, service design, governance and program delivery is fundamental to reducing stigma and discrimination, building trust and safety with clients, and improving the cultural safety and utilisation of harm reduction service. Funding should support an increase in the Aboriginal and Torres Strait Islander workforce, including paid peer roles [58,66]. Finally, the lack of healthcare providers with perceived empathy towards Aboriginal and Torres Strait Islander PWID and the issues of racism and enacted stigma urgently need to be addressed. Solutions to educate and increase mainstream healthcare workers’ empathy may include cultural competency training led by paid Aboriginal and Torres Strait Islander educators or peer workers that centres on dismantling colonial narratives and encouraging healthcare workers to critically reflect on their biases and adopt humility in their healthcare practices [2].

While interventions such as cultural competency training for healthcare workers are urgently needed, there must also be a focus on community-level interventions that address discrimination and stigmatisation at a structural level [19]. The theoretical perspective of intersectional stigma can inform the ways in which interventions are designed for Aboriginal and Torres Strait Islander PWID. This framework suggests the need for macro-level strategies, such as community-led anti-stigma campaigns, policy reform and decriminalisation, and anti-stigma media rhetoric, which address the structural sources of stigmatisation and inequity [19,67]. Reducing the harms of stigma and discrimination therefore requires multi-level and systematic approaches that draw on the strengths and resilience of communities.

### 4.2. Limitations

Our survey estimates of the proportions experiencing different types of stigma and discrimination may be biased (under- or over-estimates of community prevalence) because they are derived from a small sample recruited using convenience and chain referral approaches. Further, the personal stories of participants involved in the interview yarns are unlikely to capture the full diversity of such experiences among Aboriginal and Torres Strait Islander PWID at these regional and major city locations. In addition, in the interview yarns, we did not directly ask participants about their experiences of stigma; these experiences were either raised independently amongst the various topics and stories that were shared, or in response to questions such as “have you ever been made to feel unwelcome in a healthcare setting?”.

Although most participants spoke highly of the staff at the NSP research sites, it is important to acknowledge the lack of reported concerns may be due to participants feeling apprehensive about sharing negative views about these services in fear of retribution or affecting future access. This has been acknowledged in other studies with Aboriginal and Torres Strait Islander peoples [38]. This research did not follow culturally appropriate protocols for men’s business and women’s business [68], and all interviews were carried out by a female research assistant, and another female non-Indigenous research assistant. This was attributed to limited timeframes and project resources and may have affected our ability to build rapport. Finally, this research engaged with service-connected Aboriginal and Torres Strait Islander PWID, and those referred to the research by family and friends, which may exclude those most alienated from healthcare services. “Hard to reach” is a common narrative used to explain the difficulty of engaging and recruiting populations such as Aboriginal and Torres Strait Islander PWID [69,70], but it may be important to rephrase this narrative to emphasise health system failure and the need to make healthcare services and harm reduction programs accessible for this population.

## 5. Conclusions

Racism and stigma experienced by Aboriginal and Torres Strait Islander PWID are fundamental barriers to equitable, accessible and culturally safe healthcare and harm reduction services for these groups. Centring on Aboriginal and Torres Strait Islander peoples’ lived/living experiences in the development of harm reduction and other healthcare services is urgently needed to improve health and wellbeing outcomes. Program and policy development requires coordinated holistic and intersectoral approaches, with strong involvement of community-controlled organisations that shift the balance of power to Aboriginal and Torres Strait Islander communities. Despite the profound health inequities and social circumstances faced by these groups, and the failure of Australia’s healthcare system to provide culturally safe care, Indigenous-led approaches that are grounded in connection to Country, family, and community are possible to promote emotional and physical healing for Aboriginal and Torres Strait Islander peoples [2,71,72].

## Figures and Tables

**Table 1 ijerph-22-01120-t001:** Participants who had experienced different forms of stigma and discrimination, reporting per cent (%) and 95% confidence interval (CI) for ‘any’ and ‘high-level’ stigma/discrimination, *n* = 94.

Measure of Discrimination/Stigma	Any		High Level	
	%	95% CI	%	95% CI
Everyday discrimination ^a^	85.1	76.3–91.6	28.7	19.9–39.0
Healthcare discrimination ^a^	70.2	59.9–79.2	19.1	11.8–28.6
Enacted SU stigma—family ^b^	71.3	61.0–80.1	27.7	18.9–37.8
Enacted SU stigma—healthcare ^b^	54.3	43.7–64.6	11.7	6.0–20.0
Anticipated SU stigma—family ^b^	66.0	55.5–75.4	25.5	17.1–35.6
Anticipated SU stigma—healthcare ^b^	54.3	43.7–64.6	12.8	6.8–21.2
Internalised SU stigma ^b^	77.7	67.9–85.6	16.0	9.2–25.0
Anticipated HCV stigma ^c,d^ (*n* = 43)	58.1	42.1–73.0	14.0	5.3–27.9
Internalised HCV stigma ^c,d^ (*n* = 44)	40.9	26.3–56.8	4.5	0.5–15.5

^a^ Mayi Kuwayu Study discrimination scales; ^b^ Substance Use Stigma Mechanisms Scale (SU-SMS); ^c^ Hepatitis C Stigma Scale (HCV Stigma Scale); ^d^ there is *n* = 43 and *n* = 44 for the respective HCV stigma items because these questions only pertain to those participants who had ever been diagnosed with hepatitis C.

## Data Availability

Data is available from the authors upon reasonable request.

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
