# Peer review of "The Unique Experience of Intersectional Stigma and Racism for Aboriginal and Torres Strait Islander People Who Inject Drugs, and Its Effect on Healthcare and Harm Reduction Service Access"

_ijerph, 2025, doi:10.3390/ijerph22071120_

Round 1
Reviewer 1 Report
Comments and Suggestions for Authors
This article examines how intersectional stigma and racism influence healthcare and harm reduction access for Aboriginal and Torres Strait Islander people who inject drugs (PWID) in Queensland, Australia. Drawing on a mixed-methods design, the authors conducted a cross-sectional survey with 94 participants and a series of yarning interviews. The study identifies three central themes: 1) complex social and mental health needs limit engagement with services; 2) experiences of stigma and racism in healthcare settings deter service use; and 3) injecting drug use contributes to cultural disconnection, reinforcing shame and barriers to care. The paper offers important insights into how overlapping forms of stigma impact health equity and proposes directions for more culturally safe and accessible services.
General Comments
- This research addresses an important and under-explored issue at the intersection of Indigenous health, substance use, and stigma. The topic is highly relevant to public health and fits well within the scope of IJERPH. There has been little prior exploration of how multiple forms of stigma co-occur for Aboriginal and Torres Strait Islander PWID, so the study offers novel insights. The findings are significant for service providers and policymakers working to improve harm reduction and healthcare services for Indigenous communities.
- The overall study design is appropriate and methodologically sound. The mixed-methods approach is well-justified, providing both breadth and depth in understanding the research question. The yarning interview method is an excellent choice for building trust and yielding rich qualitative data using a Two-Eyed Seeing Approach/Indigenous knowledges.
- The quantitative component consists of a descriptive survey of stigma experiences. While basic (reporting proportions and means), these measures appropriately quantify the prevalence of discrimination in healthcare, everyday life, family/friends, and hepatitis C-related stigma. The sample size (N=94) is adequate for descriptive insights, though not for complex statistical comparisons; the authors rightly focus on frequencies and avoid over-interpretation.
- The qualitative analysis is a key strength, enhanced by the involvement of Aboriginal team members, advisors, and stakeholders in coding and theme validation. This collaborative approach demonstrates strong cultural sensitivity and ethical rigor. The use of culturally appropriate methods, such as yarning, likely fostered trust and safety among participants, and the emphasis on culturally safe care aligns with broader efforts to improve Indigenous health equity.
Specific Comments:
Introduction
- Introduce or expand discussion of Indigenous health paradigms around line 104. For example, mention the Social and Emotional Wellbeing framework or Indigenous concepts of holistic health to frame how stigma impacts not just individuals but kinship ties, cultural identity, and connection.
Methods
- Add a sentence or two on researcher reflexivity, which aligns with Braun and Clarke. For example, “The research team engaged in reflexive discussions throughout analysis, examining how our positions (Indigenous and non-Indigenous, academic and lived-experience) might influence interpretations.”
- If any culturally safe practices were employed but not mentioned, include them. For example, was there any supports or debriefing afterwards? Was there smudging or other cultural practices? Were Elders or Aboriginal advisors present to provide support?
- The methods section briefly mentions that one interviewer was Aboriginal and that the study had Aboriginal advisors. However, it does not make clear whether the project followed Indigenous governance principles (e.g., OCAP) or how community input shaped the research questions, design, or dissemination.
- There is a discrepancy in the sample counts (23 + 66 + 20 = 109), which does not match the stated total of 94 participants. Please clarify or correct this inconsistency in the text.
Results
- The manuscript would benefit from a clearer description of the sample’s characteristics. I recommend adding a short summary (either in text or a small table) of participant characteristics (e.g., mean age or age range, gender, etc.), as this context will help readers understand the sample.
- In Table 1, only percentages are reported. I recommend including the corresponding counts for clarity. Additionally, it is unclear why the number of respondents for the HCV stigma question is 43 and 44. Suggest clarifying this discrepancy in the methods section.
- Line 339: There is a typo in the quote. ‘Alight’ should be ‘all right’.
- Line 328: In the quote “you know I don’t feel ashamed” (Participant 12), the text before it is marked “(unclear)”. If the audio was unclear, consider if that segment is needed; if it’s not critical, you might omit the unclear portion for readability.
- Theme 3 could be expanded in two important ways. First, while it highlights disconnection from culture and identity, many participants also describe a desire to give back, such as mentoring youth, returning home, or reclaiming purpose. These reflections speak to relational accountability and kinship responsibilities, which are central in many Indigenous worldviews but are underexplored here. Second, the theme presents internalized stigma (e.g., shame) clearly, but could be strengthened by incorporating the concept of stigma resistance or resilience, emphasizing how individuals actively resist or reframe stigma as part of survival and healing.
Discussion
- If the authors are open to expanding their interpretive lens, the concept of intersectional resilience could help frame some of the emergent themes, such as “warrior” identities and the protection of younger generations, as acts of resistance and adaptation in the face of intersecting stigmas. While this wouldn’t require re-analysis, it could enrich the discussion and align the manuscript with strengths-based directions in intersectional qualitative research.
- It would strengthen the paper to integrate the survey findings a bit more into the discussion. For instance, the survey results showed a high prevalence of discrimination and internalized stigma (as noted in Results 3.1). The authors could explicitly mention how those quantitative findings corroborate the qualitative narratives.
- Adding a couple of sentences on practical implications would enhance the paper’s relevance for healthcare and policy audiences. For example, participants described feeling safer in peer-led spaces, suggesting a need to expand Indigenous community-led and peer-led services. Similarly, given reports of racism and disconnection, involving Elders or Aboriginal advisors in harm reduction services and strengthening cultural safety training for mainstream providers could be recommended.
- Line 502: People who inject drugs should be changed to the acronym PWID for consistency.
- The discussion could be slightly more explicit about how stigma translates to poor outcomes. For instance, the data show stigma leads to avoidance of care (participants avoid hospitals, or delay seeking help until emergencies). The authors could connect this to known consequences: untreated infections, late presentation to treatment, continued risky behaviors, etc.
- In the limitations, note that participants were service-connected PWID, which may exclude those most alienated from healthcare. Emphasize the need to reach hidden populations (possibly via outreach or RDS) in future studies to see if their stigma experiences differ.
Reviewer 2 Report
Comments and Suggestions for Authors
This mixed methods study examines how racial/ethnic stigma and stigma associated with problem substance use are experienced by Aboriginal and Torres Strait indigenous people.
The theoretical context and overal research goals are well articulated but the goals of the current paper and its particular questions and answers are implied and need to be more clearly referenced in justifying methods and data reported.
The study includes quantitative data from 94 people who are both Aboriginal/Torres Strait islanders and substance users----we see means and other descriptives for multiple measures of experiened stigma based on race/ethnicity, substance use and both....the text offers some comparisons to another study using similar measures but not focused on substance users---the significance of the observed patterns are not made clear.....and then the study spends much of the remainder of results and discussion on the qualitative interviews with 14 people......There is no use of the quantitative data to refine or clarify the multiple dimensions of stigma and racism and their interactions or relationships to life situations. Despite the small size of this group a deeper quantitative exploration of these data that is linked to a theory or pratice question would make a useful and unique contribution. The ways in which the qualitative data do and do not support the concepts expressed in terms of quantitative analysis need to be articulated? What does the qualitative add that is beyond the quantitative?
I would also like to see more of a justification for the qualitative sample.....why did you stop at 14 people, mostly men and in some cases their partners? Did you acheive notable thematic saturation at this point? How do you understand the qualitative sample to be biased compared the quantitative?
Round 2
Reviewer 2 Report
Comments and Suggestions for Authors
The authors responded to some of my comments and seem to have also gotten additional and very helpful comments. I particularly appreciated the addition of descriptive demographic information on the quantitative sample and it also provided more confidence about the representativeness/reliability of the qualitative sample.
I still think you should provide more definition of the variables displayed in Table 1 and how these rates of any and high level discrimination/stigma are best interpreted. I appreciate the comparison to the Kuwaya study in the discussion.....and still it would be helpful to explicate these dimensions and their importance in the introduction and/or methods section.
For me, one of the more compelling features of your discussion of qualitative interviews and yarns is the findings of repeated stigmatization/discrimination from multiple sources. I am just guessing that this includes persons who are not from the Torres Islands and also members of the Aboriginal and Torres Strait Islander community and maybe there are other key group signifiers like health care staff and employers. It would be helpful to clarify this: it seems to me that a range of of programmatic interventions could be offered to lessen the impacts of discrimination would be needed to address different groups---modern/old fashioned racism and as you mention, internalized racism-- and their attitudes that sustain stigmatization.
